# TTYH1 and TTYH2 Serve as LRRC8A-Independent Volume-Regulated Anion Channels in Cancer Cells

**DOI:** 10.3390/cells8060562

**Published:** 2019-06-09

**Authors:** Yeonju Bae, Ajung Kim, Chang-Hoon Cho, Donggyu Kim, Hyun-Gug Jung, Seong-Seop Kim, Jiyun Yoo, Jae-Yong Park, Eun Mi Hwang

**Affiliations:** 1College School of Biosystem and Biomedical Science of Health Science, Korea University, Seoul 02841, Korea; younju21@hanmail.net (Y.B.); chois007@korea.ac.kr (C.-H.C.); wjdjd77@kist.re.kr (H.-G.J.); mykss816@naver.com (S.-S.K.); 2Center for Functional Connectomics, Korea Institute of Science and Technology (KIST), Seoul 02792, Korea; kitkim819@kist.re.kr; 3KHU-KIST Department of Converging Science and Technology, Graduate School, Kyung Hee University, Seoul 02447, Korea; 4Namhae Garlic Research Institute, Gyeongnam 52430, Korea; donggyukim3@gmail.com; 5Division of Applied Life Science (BK21 Plus), Research Institute of Life Sciences, Gyeongsang National University, Jinju 52828, Korea; yooj@gsnu.ac.kr

**Keywords:** VRAC, TTYH1, TTYH2, LRRC8A, cancer cells

## Abstract

Volume-regulated anion channels (VRACs) are involved in cellular functions such as regulation of cell volume, proliferation, migration, and cell death. Although leucine-rich repeat–containing 8A (LRRC8A) has been characterized as a molecular component of VRACs, here we show that Drosophila melanogaster tweety homologue 1 and 2 (TTYH1 and TTYH2) are critical for VRAC currents in cancer cells. LRRC8A-independent VRAC currents were present in the gastric cancer cell line SNU-601, but almost completely absent in its cisplatin-resistant derivative SNU-601-R10 (R10). The VRAC current in R10 was partially restored by treatment with trichostatin A (TSA), a histone deacetylase inhibitor. Based on microarray expression profiling of these cells, we selected two chloride channels, TTYH1 and TTYH2, as VRAC candidates. VRAC currents were completely absent from TTYH1- and TTYH2-deficient SNU-601 cells, and were clearly restored by expression of TTYH1 or TTYH2. In addition, we examined the expression of TTYH1 or TTYH2 in several cancer cell lines and found that VRAC currents of these cells were abolished by gene silencing of TTYH1 or TTYH2. Taken together, our data clearly show that TTYH1 and TTYH2 can act as LRRC8A-independent VRACs, suggesting novel therapeutic approaches for VRACs in cancer cells.

## 1. Introduction

Volume-regulated anion channels (VRACs) are functionally expressed in almost all cell types, including cancer cells, and are intimately involved in regulation of cell volume, proliferation, migration, and death [1]. These channels open in response to osmotic swelling of the cell, and serve to restore the cell volume to its original state by releasing chloride ions and various organic osmolytes [2]. Attempts to identify the molecular components of VRACs have been ongoing for decades. In 2014, two research groups independently reported that leucine-rich repeat–containing 8A (LRRC8A) is a core component of VRAC [3,4].

Although LRRC8A has been reported to function as a VRAC in many different cell types from a wide range of tissues, several studies suggested that it may not account for all VRAC functions. For example, when LRRC8A is knocked down in HeLa cells, induced VRAC currents are diminished but still clearly present; moreover, knockdown does not affect cell death, and residual VRAC currents are still observed in LRRC8A-knockout HCT116 cells [5]. Another group showed that, in VRAC-deficient KCP-4 cells, VRAC currents are not restored by overexpression of LRRC8A alone or LRRC8A in combination with other LRRC8 isoforms [6]. In human retinal pigment epithelium (RPE) cells, bestrophin 1 (BEST1) is crucial for VRAC currents and volume regulation, and stable knockdown of LRRC8A has no effect [7]. Collectively, these studies suggested that channels that do not contain LRRC8A could also act as VRACs.

The Drosophila melanogaster tweety (tty) gene, originally isolated as a transcription unit adjacent to flightless-1, encodes a large conductance chloride channel that belongs to a highly conserved and evolutionarily ancient family [8,9]. In 2004, human homologues of tweety (TTYH1–3) were identified and characterized: TTYH1 is a swelling-activated chloride channel, and TTYH2 and TTYH3 act as calcium-dependent maxi-chlorine channels when overexpressed in Chinese hamster ovary (CHO) cells [10]. To date, however, no study has examined VRAC activity in cells endogenously expressing TTYH1.

In contrast to cisplatin-sensitive cancer cells, cisplatin-resistant cancer cells do not have VRAC currents in hypotonic solution [11,12]. In addition, when cisplatin-resistant cells are treated with trichostatin A (TSA), a histone deacetylase inhibitor, the VRAC currents are partially restored. Based on these observations, we predicted that the expression of genes responsible for VRAC activity should be altered in these cancer cells. Hence, to identify the gene(s) responsible for VRAC, we screened VRAC currents in several cancer cell lines and their cisplatin-resistant derivatives. Specifically, we examined the gastric cancer cell line SNU-601 and its cisplatin-resistant derivative SNU-601/Cis10 (R10). VRAC activity was absent in R10, but was restored by TSA treatment. Based on the gene expression profiles of these three types of cells (VRAC-active, VRAC-deficient, and VRAC-restored), we selected two candidate VRAC genes, TTYH1 and TTYH2, and confirmed that the channels they encode can act as VRACs in several types of cancer cells.

## 2. Materials and Methods

### 2.1. Cell Culture

SNU-601, SNU-601/Cis10 (R10), and LoVo cells were grown in RPMI 1640 (Thermo Fisher Scientific, Waltham, MA, USA). HEK293T, HepG2, and MCF-7 cells were cultured in DMEM (Thermo Fisher Scientific). Culture medium was supplemented with 10% fetal bovine serum (Thermo Fisher Scientific) and 1% penicillin–streptomycin, and cells were grown at 37 °C in a 5% CO_2_ incubator. The TTYH1 and TTYH2 double-knockout derivative of SNU-601 (dKO) was generated by Korea Bio (Seoul, Korea) using the CRISPR/Cas9 system (Santa Cruz, CA, USA). Knockouts of genes were confirmed by PCR and sequencing.

### 2.2. Microarray and Analysis

Microarray analyses were performed using a commercial microarray service (Ebiogen, Seoul, Korea). Total RNA was isolated from SNU-601, R10, and R10_TSA cells using the RNA Purification Kit (GeneAll, Seoul, Korea) and subjected to cDNA microarray analyses, performed by eBiogene (Seoul, Korea). Data normalization was performed using the GeneSpringGX7.3 software (Agilent Technology, Santa Clara, CA, USA). Briefly, averages of normalized ratios were calculated by dividing normalized test channel intensities by normalized control channel intensities. Assessment of functional annotations and Gene Ontology (GO) was performed using GeneCards (https://www.genecards.org) web-based analysis.

### 2.3. Immunocytochemistry

SNU-601 and dKO cells were grown on poly-D-lysine-coated coverslips. After washes in PBS, cells were fixed in 4% paraformaldehyde in PBS for 15 min at room temperature, and then rinsed three times with PBS. Cells were permeabilized with 0.2% Triton X-100 in PBS for 7 min, and then incubated in blocking buffer (5% normal donkey serum, 3% BSA and 0.1% Triton X-100 in PBS) for 1 h. The cells were incubated with anti-TTYH2 antibody (Invitrogen, Carlsbad, CA, USA, PA5-34395) at 4 °C overnight. The next day, the cells were washed three times in PBS and incubated with suitable Alexa Fluor-tagged secondary antibodies (Jackson ImmunoResearch, West Grove, PA, USA). After incubation with secondary antibodies, the cells were treated with phalloidin at room temperature for 30 min to label the plasma membrane. The coverslips were mounted on slides, and the cells were imaged by confocal microscopy (A1 confocal microscope; Nikon, Tokyo, Japan).

### 2.4. shRNA Construction and Validation

Short hairpin shRNA (shRNA) vectors for human LRRC8A, TTYH1, and TTYH2 were constructed using pSicoR (Addgene, #11579) with GFP replaced by mCherry. shRNA sequences were as follows: LRRC8A: 5′-GGTACAACCACATCGCCTA-3′ [3]; TTYH1: 5′-GCATCGGTTTCTATGGCAACA-3′; TTYH2: 5′-GGATTATCTGGACGCTCTTGC-3′ [13]. A pSicoR construct containing a scrambled shRNA was used as a control. To validate shRNAs, SNU-601, HEK293T, HepG2, and LoVo cells were cultured and transfected with each shRNA in the presence of Lipofectamine 2000 (Invitrogen) in serum-free culture medium for 6 h, and then incubated in normal culture medium for 72 h. The efficiency of gene silencing was assessed by RT-PCR and western blot.

### 2.5. Western Blot

For western blotting, HEK293T, SNU-601, and TTYH1/2 double-knockout cells were lysed with lysis buffer (50 mM HEPES, 0.1% sodium deoxycholate, 1% Triton X-100, 1 mM PMSF, and 0.1% SDS containing protease inhibitor cocktail, pH 7.4). Total protein (15 μg/lane) was subjected to SDS-PAGE (8–12% gels) and transferred to PVDF membranes. The membranes were blocked using 5% non-fat milk, and then blotted with the appropriate antibodies: Anti-LRRC8A (Cell Signaling Technology, Inc., Danvers, MA, USA, #24979), anti-TTYH1 (CUSABIO technology LLC, Houston, TX, USA, CSB-PA867139LA01HU), and anti-actin (Sigma, St. Louis, MO, USA, A5441). The membranes were then washed and incubated with HRP-conjugated goat anti-mouse (Jackson ImmunoResearch, #115-035-166), goat anti-rabbit (Jackson ImmunoResearch, #111-035-144), or rabbit anti-goat IgG (Jackson ImmunoResearch Lab, #305-035-045), followed by washing and detection of immunoreactivity using ECL (Thermo, #34095).

### 2.6. RT-PCR and Real-Time PCR

Total RNA was isolated from cells using the RNA Purification Kit from GeneAll. cDNAs were synthesized from 1 μg total RNA, and reverse transcription was performed using the SensiFAST™ cDNA Synthesis Kit (BIOLINE, London, UK). RT-PCR primer sequences were as follows: LRRC8A: Forward, 5′-CTTCTCCTGAGTTCCTGGTC-3′; reverse, 5′-AAGGATGGCTCTGCTATCTG-3′; TTYH1: Forward, 5′-CTGGTGATCGTGATGACAGT-3′; reverse, 5′-TGCACCATAGTCCTTGTGCA-3′; TTYH2: Forward, 5′-GTGGACTACATCGCTCCCTG-3′; reverse, 5′-TGAACTTCAGGGTCTGCAGG-3′; and GAPDH: Forward, 5′-GTCTTCACCACCATGGAGAA-3′; reverse, 5′-GCATGGACTGTGGTCATGAG-3′. GAPDH was used as a loading control. The LRRC8A, TTYH1 and TTYH2 fragments were amplified under the following cycle conditions: Denaturation at 94 °C for 30 s, annealing at 57 °C for 30 s, and extension at 72 °C for 30 s. This cycle was repeated a total of 32 times. GAPDH fragments were also amplified under the same conditions except that 24 cycles were run. Real-time PCR was performed using the SensiFAST™ Probe Hi-ROX kit (Invitrogen). Primer sets for LRRB8A (Hs.PT.58.346894), LRRC8B (Hs.PT.58.12310), LRRC8C (Hs.PT.58.1804454), LRRC8D (Hs.PT.58.15346618), LRRC8E (Hs.PT.58.48538734), TTYH1 (Hs.PT.58.40149255), TTYH2 (Hs.PT.58.4787185), TTYH3 (Hs.PT.58.21128423), CFTR (Hs.PT.58.3365414), and GAPDH (Hs.PT.39a.22214836) were generated by IDT (PrimeTime qPCR assays). All experiments were repeated at least three times. The 2^-ΔΔC^_T_ method was used to calculate fold changes in gene expression [14].

### 2.7. Electrophysiology

The standard pipette solution contained the following (in mM): 60 Trizma-HCl, 70 Trizma-base, 70 aspartic acid, 15 HEPES, 0.4 CaCl_2_, 1 MgCl_2_, 1 EGTA, 1 ATP, and 0.5 GTP; the pH was adjusted to 7.2 with CsOH. The bath contained the following (in mM): 70 Trizma-HCl, 1.5 CaCl_2_, 10 HEPES, 10 D-glucose, and 100 sucrose (290 mOsm/kg); the pH was adjusted to 7.4 with CsOH. To block K^+^ currents, 5 mM TEA and 5 mM BaCl_2_ were added to the bath solution. Hypotonic solution had the same ionic composition as the bath solution but lacked sucrose (220 mOsm/kg; the pH was adjusted to 7.4 with CsOH). Patch pipettes were made from borosilicate glass capillaries (Warner Instruments, Hamden, CT, USA), and had a resistance of 5–6 MΩ. Whole-cell currents were recorded using a patch-clamp amplifier (Axopatch 700B; Molecular Devices, Silicon Valley, CA, USA). Current–voltage relationships were measured by applying ramp pulses (from −100 mV to +100 mV, 1 s duration) from a holding potential of −60 mV. A Digidata 1550A interface was used to convert digital–analogue signals. Data were sampled at 5 kHz and filtered at 1 kHz. Currents were analyzed with the Clampfit software (Molecular Devices). All experiments were conducted at room temperature.

### 2.8. Statistical Analysis

All statistical analyses were performed using GraphPad Prism version 8.00 (GraphPad Software, La Jolla, CA, USA) for Windows. Numerical data are presented as means ± standard error of the mean (SEM). Statistical differences were evaluated by ANOVA and the paired or unpaired Student’s *t*-test, as appropriate. Differences were considered significant at *P* < 0.05.

## 3. Results

### 3.1. VRAC Currents are Shown in SNU-601 Cells but not in Cisplatin-Resistant R10 cells

To observe VRAC activity, we used whole-cell patch-clamp recording in the gastric cancer cell line SNU-601 and its cisplatin-resistant derivative SNU-601/Cis10 (R10). R10 cells were generated by chronic exposure to 10 mg/mL cisplatin, a platinum-containing anti-cancer drug [15]. In hypotonic solution, VRAC-like currents were gradually induced in SNU-601 cells that were similar to those observed in other cancer cells [11], but no current was detected in R10 cells. In addition, the current–voltage (*I–V*) relationship indicated that the I_Cl_ currents in SNU-601 cells were dramatically increased in response to hypotonic solution. However, the *I–V* relationship of I_Cl_ currents remained almost unchanged in R10 cells (Figure 1b,c). To determine whether the hypotonicity-induced I_Cl_ currents in SNU-601 cells were VRAC currents, we treated cells with DCPIB, a selective blocker of VRAC [16,17]. The elevated I_Cl_ currents in hypotonic solution were inhibited in 30 μM DCPIB (Figure 1d,e). These findings suggest that SNU-601 gastric cancer cells have volume-regulated I_Cl_ currents, whereas cisplatin-resistant R10 cells do not.

### 3.2. SNU-601 Cells Have LRRC8A-Independent VRAC Currents

Previous studies showed that LRRC8A (SWELL1) is a key component of the VRAC [3,4]. Therefore, we first investigated whether the hypotonicity-induced I_Cl_ currents in SNU-601 cells were dependent on LRRC8A. To this end, we constructed a shRNA against LRRC8A and confirmed that it efficiently silenced LRRC8A expression in SNU-601 cells (Appendix A). In SNU-601 cells transfected with LRRC8A shRNA, hypotonicity-induced VRAC currents were comparable to those in SNU-601 cells transfected with control scrambled shRNA (Figure 2a,b). Because this result was unexpected, we examined VRAC currents in HEK293T cells, in which LRRC8A was originally identified as a VRAC component [3]. In HEK293 cells transfected with LRRC8A shRNA, VRAC currents were not induced in hypotonic solution, as previously reported (Figure 2c,d).

Because LRRC8A has four closely related homologues (LRRC8B–E) and forms heteromers [4,18], we examined the expression levels of the five LRRC8 family members in SNU-601 and R10 cells by quantitative RT-PCR (qRT-PCR) (Figure 2e). Relative expression levels of LRRC8A, LRRC8D, and LRRC8E were unchanged between SNU-601 and R10 cells, but LRRC8B was higher in R10. These data suggested that expression of the other LRRC8 family members was not correlated with cisplatin resistance in these cells. Taken together, these findings indicate that hypotonicity-induced VRAC currents in SNU-601 cells do not depend on LRRC8A expression.

### 3.3. Whole-Genome Screening Identifies TTYH1 and TTYH2 as VRAC Candidates

TSA is a histone deacetylase inhibitor and potent anti-cancer agent [19]. A previous study showed that TSA partially restored VRAC currents in KCP-4 cells [11]. Therefore, we investigated whether TSA treatment could restore VRAC currents in cisplatin-resistant R10 cells. As shown in Figure 1a, VRAC currents in R10 cells were not induced in hypertonic solution, but were restored in R10 cells treated with TSA (Figure 3a). To identify candidate genes encoding VRAC components or their regulators in SNU-601 cells, we used Whole Human Genome Microarrays (44,000 genes) to monitor expression levels in SNU-601, R10, and TSA-treated R10 cells. We initially focused on proteins predicted to have at least one transmembrane (TM) domain (6081 genes) (Figure 3b). From this list, we identified chloride channels (95 genes, Appendix A), and then selected three candidate genes (TTYH1, TTYH2, and CFTR) that were expressed in R10 cells at 20% of the level in SNU-601 cells. Of those, only two (TTYH1 and TTYH2) were more than 2-fold upregulated in TSA-treated R10 cells relative to untreated R10 cells (Figure 3b).

Initially, we generated scatter plots of log (read numbers) in SNU-601 cells vs. R10 cells (Figure 3c, left). This analysis confirmed that TTYH1, TTYH2, and CFTR genes were down-regulated in R10 cells. Next, we compared the scatter plots representing the log (read numbers) in R10 cells vs. TSA-treated R10 cells (Figure 3c, right), and found that the level of CFTR expression was still low in TSA-treated R10 cells. In addition, we confirmed the relative levels of TTYH1, TTYH2, and CFTR in three different cell groups using qRT-PCR (Figure 3e). The mRNA levels of TTYH1 and TTYH2 were significantly reduced in R10 cells, but recovered in TSA-treated R10 cells. In the case of CFTR, the mRNA level was reduced in R10 cells, but did not recover in TSA-treated R10 cells. Expression of TTYH3, another TTYH family member, was highest in R10 cells, indicating that TTYH3 may not function as a VRAC (Appendix A). Together, these results identified TTYH1 and TTYH2 as a VRAC candidate.

### 3.4. TTYH1/TTYH2 dKO Cells Lack VRAC Currents

To further investigate whether TTYH1 and TTYH2 are responsible for hypotonicity-induced VRAC currents, we used TTYH1/TTYH2 double-knockout (dKO) cells generated by CRISPR/Cas9 gene editing. Exon 7 of TTYH1 and exons 2 and 3 of TTYH2 were targeted, and the double knockout was verified by western blotting. As shown in Figure 4a, expression of TTYH1 was abolished in dKO cells, whereas LRRC8A expression was unchanged. In immunocytochemical staining, TTYH2 was not detected in the dKO cells (Figure 4b). To determine whether dKO cells had VRAC activity, we measured hypotonicity-induced VRAC currents in dKO cells transfected with GFP control, GFP-tagged TTYH1, and GFP-tagged TTYH2 (Figure 4c–g). In the GFP control–transfected dKO cells, VRAC currents were not induced in hypotonic solution (Figure 4c). By contrast, in dKO cells transfected with GFP-tagged TTYH1 or GFP-tagged TTYH2, VRAC currents were efficiently produced under hypotonic conditions (Figure 4d,e). However, we observed no additive effects in dKO cells co-transfected with both TTYH1 and TTYH2 (Figure 4f). Together, these data indicated that both TTYH1 and TTYH2 can serve as VRACs in SNU-601 cells.

### 3.5. TTYH1 and TTYH2 Have Independent VRAC Activity in Other Cancer Cell Lines

To establish the VRAC activity of TTYH1 and TTYH2, we investigated other cancer cell lines that endogenously express these two channels. Among the cell lines we screened, HepG2 cells expressed TTYH1, but not TTYH2, whereas LoVo cells exhibited the opposite expression pattern (Figure 5a), and MCF-7 cells expressed neither mRNA. Next, we asked whether single expression of the TTYH isoform would be sufficient for hypotonicity-induced VRAC activity in each cell type (Figure 5b–e). VRAC currents were induced in hypotonic solution in both HepG2 and LoVo cells, and were almost inhibited by 30 μM DCPIB (Figure 5b–e). On the other hand, MCF-7 cells had very small VRAC currents in hypotonic solution, and these were suppressed by DCPIB (Figure 5f,g).

We next investigated the effect of TTYH1 or TTYH2 silencing on VRAC currents in HepG2 and LoVo cells. The level of TTYH1 mRNA was significantly reduced by TTYH1 shRNA in HepG2 cells (Figure 6a). The VRAC currents induced in the hypotonic solution were dramatically decreased in HepG2 cells transfected with TTYH1 shRNA relative to those in cells treated with scrambled shRNA (Figure 6b,c). Next, we examined the effect of TTYH2 shRNA in LoVo cells. TTYH2 shRNA effectively decreased the expression level of TTYH2 relative to that in cells transfected with scrambled shRNA (Figure 6d), and suppressed most VRAC currents (Figure 6e,f). Taken together, these results suggest that TTYH1 and TTYH2 function independently as VRACs in various cancer cells.

## 4. Discussion

VRACs play important roles in regulation of cell volume, proliferation, migration, and death [1]. In particular, VRACs are also involved in the responsiveness of cancers to Pt-based anti-cancer drugs such as cisplatin and carboplatin [20]. LRRC8A acts as a core component of VRAC in diverse cell types [3,4], but VRAC activity is still present in LRRC8A-knockout HCT116 and LRRC8A-knockdown HeLa cells [5]. These data raise the possibility that other channels can also act as VRACs.

In this study, we found that the VRAC currents of SNU-601 cells were largely unaffected by silencing of LRRC8A, even though this gene is efficiently expressed in SNU-601 cells (Figure 2). Consistent with a previous study of RPE cells [7], our data confirmed that LRRC8A-independent VRAC currents can be detected in this cancer cell line. In addition, the VRAC currents were almost completely abolished in cisplatin-resistant R10 cells, but were partially restored upon TSA treatment (Figure 1 and Figure 3). We used SNU-601, R10, and TSA-treated R10 cells to identify the LRRC8A-independent VRAC(s).

Our strategy for finding new VRACs was based on the reasoning that, if a specific gene encoding a LRRC8A-independent VRAC exists, its expression should be maintained in SNU-601 cells, suppressed in R10 cells, and recovered in TSA-treated R10 cells. Normally, when we compare changes in gene expression, each gene is classified into one of three groups (increase, no change, or decrease). However, if changes in gene expression are considered under three different conditions, each gene must be assigned to one of nine groups; consequently, only a small number of candidates can be selected. Based on microarray expression profiles of SNU-601, R10, and TSA-treated R10 cells, we defined a short list of candidate genes whose expression levels were correlated with VRAC currents in these three cell types (Figure 3).

Our microarray data also clearly showed that not all LRRC8 isoforms’ expression patterns matched VRAC activities in three different cells (VRAC-active, VRAC-deficient, and VRAC-restored). Likewise, the levels of mRNAs encoding other chloride channels, such as CFTR and TTYH3, also did not match (Appendix A). Among the TTYH family, only TTYH1 and TTYH2 were well correlated with VRAC activity. As shown by Figure 4, overexpression of TTYH1 or TTYH2 rescued VRAC currents in dKO cells. In addition, gene silencing of endogenous TTYH1 or TTYH2 decreased VRAC currents in HepG2 and LoVo cells, respectively. These data strongly indicate that both TTYH1 and TTYH2 act as VRACs in various cancer cells.

VRAC currents are almost absent in cisplatin-resistant KCP-4 cells [11]. Consistent with this, we found that cisplatin-resistant R10 cells almost completely lacked VRAC currents (Figure 1). These data suggest that reduced VRAC currents in cancer cells might be correlated with cisplatin resistance. We found that TTYH1 or TTYH2 could act as VRACs in cancer cells (SNU-601, HepG2, and LoVo cells), and that these channels were not expressed in MCF-7 cells (Figure 5). Notably in this regard, LoVo cells expressing TTYH2 are much more sensitive to cisplatin (IC_50,_ 0.8 µM) than MCF-7 cells (IC_50,_ 15 µM), which poorly express TTYH1 and TTYH2 [21,22]. Therefore, it is worth investigating whether gene deficiency of TTYH1 or TTYH2 is associated with the cisplatin resistance in these cancer cells. In addition, TTYH1 and TTYH2 are upregulated in glioblastoma cells and colon cancer cells, respectively [23,24]. In these cancer cells, TTYH1 and TTYH2 regulate cell proliferation and migration, which can be effectively inhibited by silencing the corresponding gene [24]. Therefore, the functional roles of TTYH1- or TTYH2-mediated VRAC currents should be examined in these cancer cells.

Interestingly, a recent report demonstrated that even though total LRRC8A expression in cisplatin-resistant A549 cells (A549CisR10) was increased compared with wild type A549 cells, taurine releasing activity of LRRC8A was reduced by the deficiency of LRRC8A surface expression [25]. Our present study also showed that mRNAs and proteins of LRRC8A were expressed in SNU-601 cells and R10 cells, but the VRAC currents were not affected by gene silencing of LRRC8A (Figure 2 and Figure 4). Therefore, it is possible that the absence of LRRC8A-dependent VRAC currents could be mediated by the impairment of LRRC8A surface expression in SNU-601 cells. The regulatory mechanism of LRRC8A surface expression should be further investigated.

In general, the functions and surface expression of ion channels are regulated by protein–protein interactions [26,27]. Interestingly, recent studies showed that TTYH1 can regulate Notch signaling in neural stem cells via protein–protein interaction with Rer1 (retention in endoplasmic reticulum sorting receptor 1), and surface expression of TTYH2 is suppressed by protein–protein interactions with β-COP [13,28]. In addition, the cellular level of TTYH2 is regulated by Nedd4-2–mediated ubiquitination [29]. These binding partners of TTYH1 or TTYH2 may be involved in the regulatory mechanisms of VRACs in diverse normal tissues and cancer cells. Furthermore, LRRC8A and TMEM16A could be associated and cell swelling can stimulate the activities of LRRC8A and TMEM16A [30]. This finding implied that LRRC8A might interact with other ion channels. Therefore, we could not exclude the possibility of the functional interaction between TTYH channels and LRRC8A, and future studies are needed to test the possibility.

Our results suggest that TTYH1 and TTYH2 can function as LRRC8A-independent VRACs. Further studies are needed to determine whether they are involved in other functions of VRACs, such as apoptosis and volume control in cancer cells. In particular, elucidation of the VRAC functions of TTYH1 as a VRAC could lead to improved treatment of brain tumors, which are currently associated with high mortality rates and limited therapeutic options.

## Figures and Tables

**Figure 1 cells-08-00562-f001:**
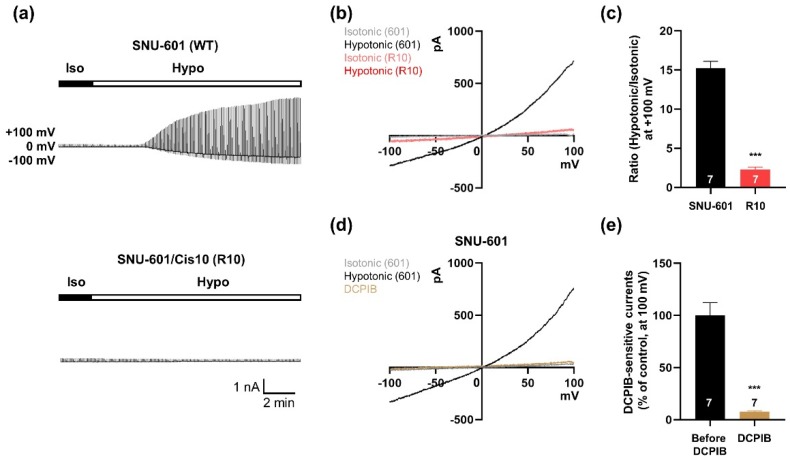
Volume-activated chloride currents in SNU-601 cells. (**a**) Representative traces showing time courses of the volume-activated chloride current in SNU-601 and R10 cells elicited by voltage ramp from −100 to +100 mV. (**b**) Representative traces showing the current–voltage relationship for volume-activated chloride currents in SNU-601 and R10 cells before and during perfusion with hypotonic solution, respectively. (**c**) Summary bar graph showing the ratio of current amplitudes of SNU-601 (n = 7) and R10 cells (n = 7) before and during perfusion with a hypotonic solution. (**d**) Representative traces of volume-regulated anion channel (VRAC) currents of SNU-601 cells before and during perfusion with a hypotonic solution, and during DCPIB application in a hypotonic solution. (**e**) Summary bar graph showing the ratio of current amplitudes of DCPIB-sensitive currents before and after DCPIB application (n = 7). Data are presented as means ± SEM (*** *P* < 0.001).

**Figure 2 cells-08-00562-f002:**
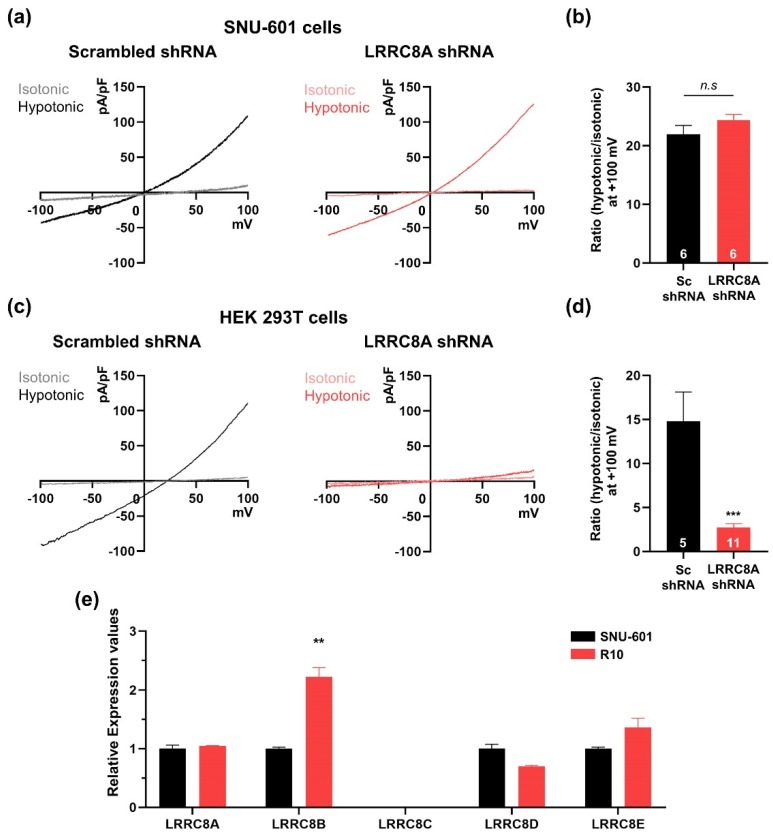
SNU-601 cells have a LRRC8A-independent VRAC activity. (**a**) Representative traces showing the current–voltage relationship for VRACs in SNU-601 cells transfected with scrambled or LRRC8A shRNAs under isotonic or hypotonic conditions. (**b**) Summary bar graph showing the ratio of current amplitudes of hypotonic/isotonic solutions in SNU-601 cells transfected with scrambled or LRRC8A shRNAs (n = 6). (**c**) Representative traces showing the current–voltage relationship for VRACs in HEK293T cells transfected with scrambled or LRRC8A shRNAs under isotonic or hypotonic conditions. (**d**) Summary bar graph showing the ratio of current amplitudes of hypotonic/isotonic solutions in SNU-601 cells transfected with scrambled shRNA (n = 5) or LRRC8A shRNA (n = 11). (**e**) Real-time PCR quantification of fold changes in LRRC8 family mRNAs in SNU-601 and R10 cells. The experiments were repeated three times. Data are presented as means ± SEM (** *P* < 0.01, *** *P* < 0.001, n.s, not significant).

**Figure 3 cells-08-00562-f003:**
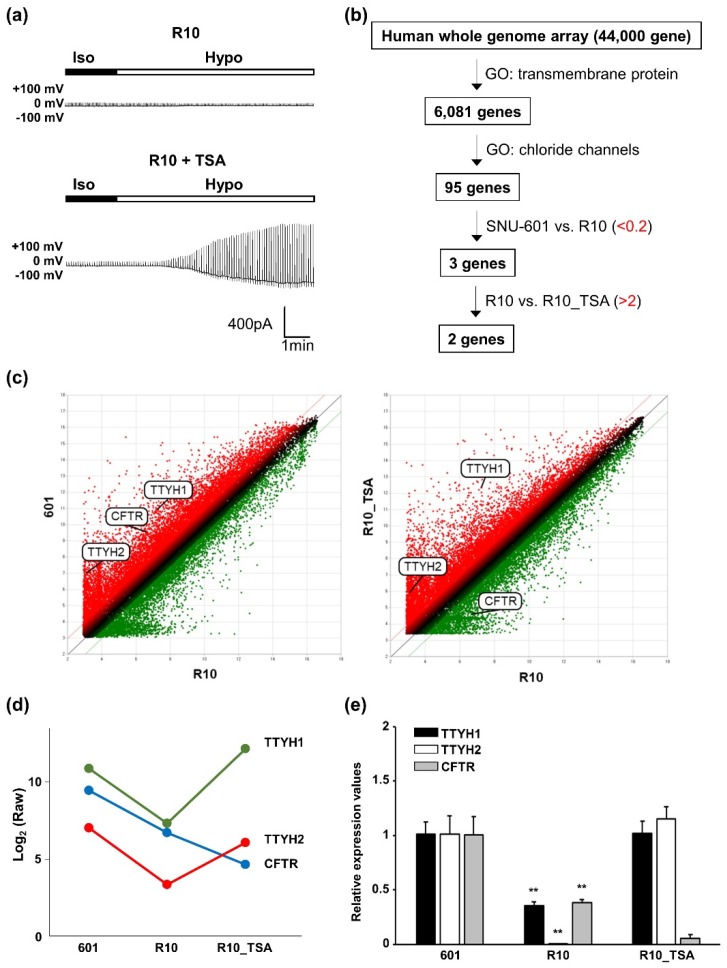
TTYH1 and TTYH2 are candidate VRACs in SNU-601 cells. (**a**) Representative traces showing the time course of volume-activated chloride currents elicited by alternating pulses by voltage ramp from −100 to +100 mV, in R10 and TSA-treated R10 cells. (**b**) Schematic overview of our human whole-genome array screen to identify candidate VRAC components. (**c**) Scatter plots of log2 (Raw) for every gene in SNU-601 vs. R10 and TSA-treated R10 vs. R10. (**d**) Dot plots of expression levels of TTYH1, TTYH2, and CFTR in SNU-601, R10, and TSA-treated R10 cells samples, calculated rom array data. (**e**) Real-time PCR quantification of fold changes in TTYH1, TTYH2, and CFTR mRNAs in SNU-601, R10, and TSA-treated R10 cells. Data are presented as means ± SEM (** *P* < 0.01).

**Figure 4 cells-08-00562-f004:**
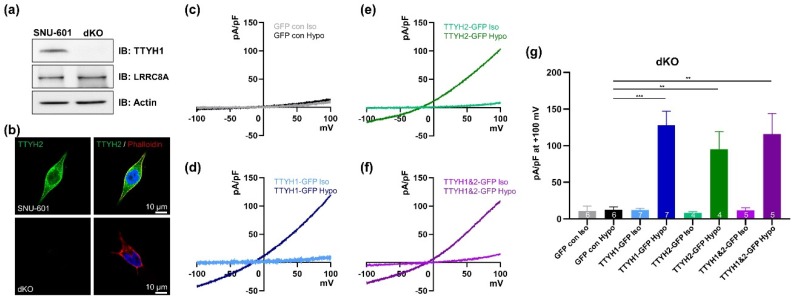
VRAC activity is completely abolished in dKO cells, and is restored by expression of TTYH1 or TTYH2. (**a**) Western blot data for TTYH1 and LRRC8A (SWELL1) in SNU-601 and TTYH1 and TTYH2 double-knockout (dKO) cells. (**b**) Representative immunocytochemical images of SNU-601 and dKO cells stained with anti-TTYH2 antibody. (**c**) Representative traces of VRAC currents from dKO cells transfected with GFP controls in isotonic and hypotonic solutions. Notably, VRAC currents were not induced by hypotonic stimulation in these cells. (**d**) Representative traces of VRAC currents from dKO cells transfected with TTYH1-GFP in isotonic and hypotonic solutions. (**e**) Representative traces of VRAC currents from dKO cells transfected with TTYH2-GFP in isotonic and hypotonic solutions. (**f**) Representative traces of VRAC currents from dKO cells co-transfected with TTYH1-GFP and TTYH2-GFP in isotonic and hypotonic solutions. (**g**) Summary bar graph showing current densities in isotonic and hypotonic solutions, as in (**c**)–(**f**), at +100 mV. Data are presented as means ± SEM (** *P* < 0.01 and *** *P* < 0.001).

**Figure 5 cells-08-00562-f005:**
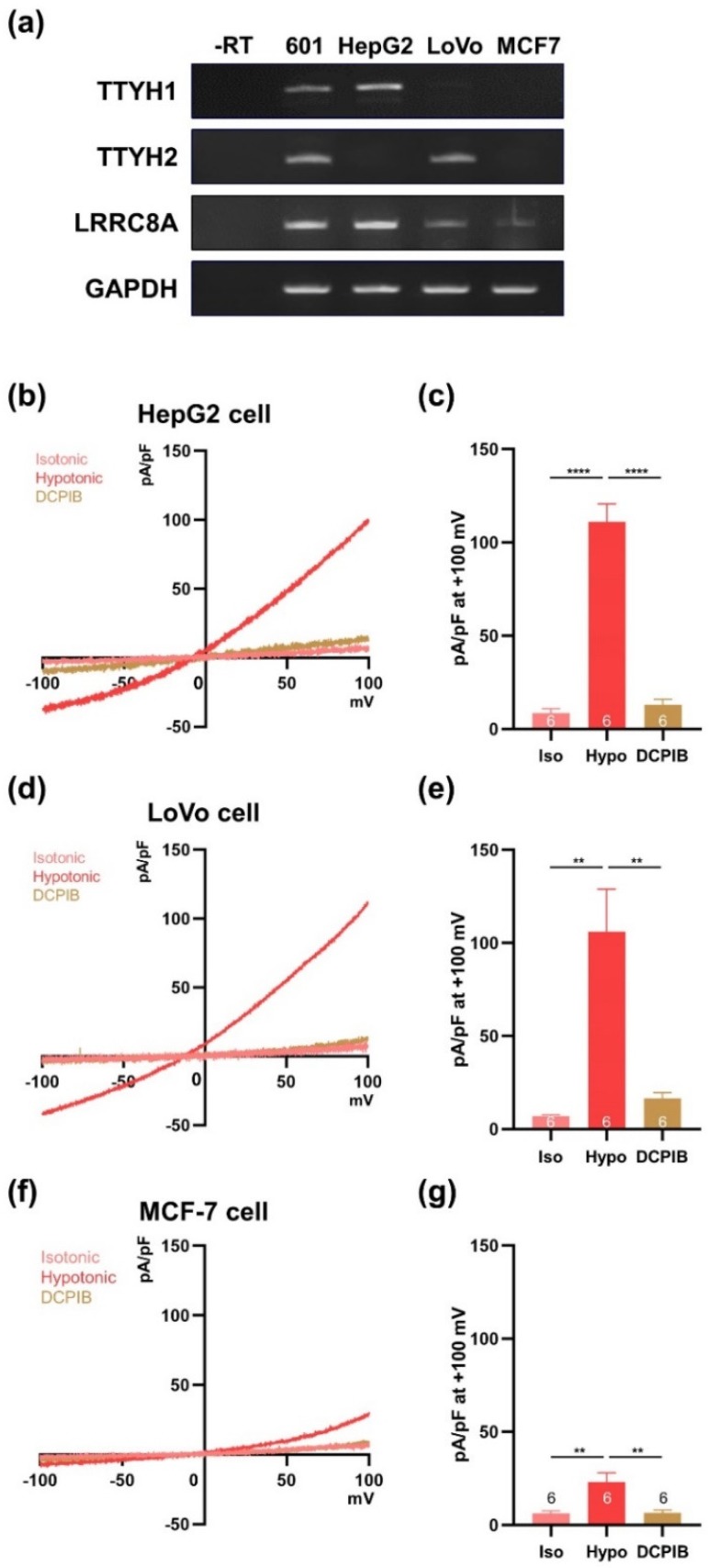
VRAC currents are observed in cancer cell lines expressing TTYH1 or TTYH2. (**a**) Expression of TTYH1 and TTYH2 mRNA in various cancer cell lines was measured by RT-PCR. (**b**) Representative traces of VRAC currents from HepG2 cells in isotonic and hypotonic solutions. (**c**) Summary bar graph showing current densities of HepG2 cells in isotonic and hypotonic solutions, as in (**b**), at +100 mV. (**d**) Representative traces of VRAC currents from LoVo cells in isotonic and hypotonic solutions. (**e**) Summary bar graph showing current densities of LoVo cells in isotonic and hypotonic solutions, as in (**d**), at +100 mV. (**f**) Representative traces of VRAC currents from MCF-7 cells in isotonic and hypotonic solutions. (**g**) Summary bar graph showing current densities of MCF-7 cells in isotonic and hypotonic solutions, as in (**f**), at +100 mV. Data are presented as means ± SEM (* *P* < 0.05, ** *P* < 0.01, **** *P* < 0.0001).

**Figure 6 cells-08-00562-f006:**
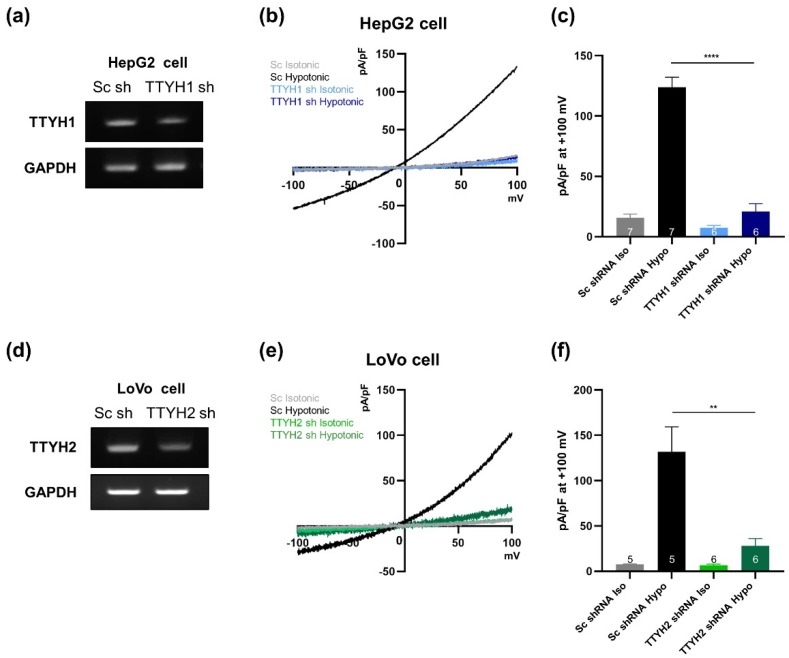
Lack of TTYH1 or TTYH2 alone in HepG2 or LoVo cells almost completely eliminates VRAC activity. (**a**) Efficiency of gene silencing by the TTYH1 shRNA construct was monitored by RT-PCR in HepG2 cells. (**b**) Representative traces of VRAC currents from HepG2 cells transfected either with scrambled or TTYH1 shRNA in isotonic and hypotonic solutions. (**c**) Summary bar graph showing current densities of transfected HepG2 cells in isotonic and hypotonic solutions, as in (**b**), at +100 mV. (**d**) Efficiency of gene silencing by the TTYH2 shRNA construct was monitored by RT-PCR. (**e**) Representative traces of VRAC currents from LoVo cells transfected with scrambled or TTYH2 shRNA in isotonic and hypotonic solutions. (**f**) Summary bar graph showing current densities of transfected LoVo cells in isotonic and hypotonic solutions, as in (**e**), at +100 mV. Data are presented as means ± SEM (** *P* < 0.01, **** *P* < 0.0001).

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
