# Peer review of "TTYH1 and TTYH2 Serve as LRRC8A-Independent Volume-Regulated Anion Channels in Cancer Cells"

_cells, 2019, doi:10.3390/cells8060562_

Round 1
Reviewer 1 Report
This is an interesting manuscript that uncovers unusual molecular nature of volume-regulated anion channel (VRAC) in several types of cancer cells. VRACs are the ubiquitous swelling-activated chloride/anion channels that mediate cell volume control, but also regulate the processes of proliferation, migration/invasion, and apoptosis. Over the past five years, numerous reports identified VRACs as the heteromers of proteins belonging to the leucine-reach repeat-containing family 8 (LRRC8). However, there are few known exceptions in which downregulation of LRRC8 expression in select cell types produced no effects on VRAC currents. Here, the Authors provide a strong evidence that in several types of cancer cells VRAC is formed by the Tweety homologue proteins TTYH1 and TTYH2. Overall, this study makes an impression and contains numerous essential controls.
Unfortunately, the manuscript needs a number of improvements, including clarification and/or update of methodology and correction of significant typos and textual errors. My specific suggestions are enumerated below.
Moderate-to-major concerns:
[1] More information is needed on cell shRNA transfection and its validation. Typically, plasmid transfection with Lipofectamine transduces not more than 50% of cells in a given culture. Consequently, UNLESS cells are antibiotic-selected, qRT-PCR and western blot analyses strongly underestimate the levels of mRNA/protein downregulation because the results are obscured by mRNA/protein from non-transfected cells. In this context, the validation data presented in Supplemental figure 1 appear to be impossible (unless cells were additionally selected, but this is not stated in Methods).
[2] The Authors state that they used real-time RT-PCR for quantification of gene expression. Yet, they present he results of only semi-quantitative RT-PCR experiments performed as visualization of gel-separated PCR products after limited number of PCR cycles (typically 30-35). There are three problems with that: (a) No qRT-PCR data are included in the manuscript; (b) The presented RT-PCR data are qualitative rather than quantitative; (b) There is no information in the text describing the presented approach (number of PCR cycles, etc.).
[3] Figure 4 legend is incorrect (it is copy-pasted from Figure 3).
[4] It is very peculiar that SNU-601 cells express significant levels of LRRC8A protein (WB in Figure in 4a) and its homologues (mRNA in Figure 2), but have no functional VRAC currents. I think that the Authors need to discuss this caveat and a potential of functional interaction between LRRC8 and Tweety proteins.
Minor concerns:
[5] In the description of electrophysiological experiments the erroneous concentration of sucrose is provided for bath solution: it does not match the described osmolarity.
[6] In Statistical Analysis (p.4) significance values should be p<0.05.< p="">
Author Response
Thank you for your comments. Please see the attached rebuttal.

Reviewer 2 Report
The manuscript by Ju Bae et al., is a very interesting basic science study which explores the expression in several cancer lines and modulation of TTYH1 and TTYH2 anion channels. The data obtained suggest these channels as novel potential target in cancer cells.
The manuscript is informative, adequately written and referenced; the experiments appear to be carefully done and they are nicely described. The quality of the figures is good.
I have no hesitation in recommending that it be accepted for publication after a few typos have been attended to:
Line 71: CO2 should be CO2
Lines 137, 138 and 140: Please correct CaCl2, MgCl2, CaCl2 and BaCl2
Line 298: Please correct: …..(Figure 6a), The VRAC currents induced
Line 352.353: “IC50” should be IC50.
Author Response

(The authors gave the same response as above.)
